

# Active and passive procrastination in terms of temperament and character

Ada H. Zohar[1,2], Lior Pesah Shimone[1] and Meirav Hen[3]

[1] Graduate Program in Clinical Psychology, Ruppin Academic Center, , Emek Hefer, Israel
[2] Lior Zfaty Center for the Prevention of Suicide and Mental Pain, Emek Hefer, Israel
[3] Psychology, Tel-Hai College, Upper Gallillee, Israel

## ABSTRACT

**Background**. While passive procrastination is usually associated with distress and dysfunction active procrastination may be an effective coping style. To test this possibility, we examined passive and active procrastination in terms of temperament, character, and emotional intelligence (EI), as well as by a short-term longitudinal study.

**Methods**. Adult community volunteers ($N = 126$) self-reported twice in an online short-term longitudinal study. At baseline on active and passive procrastination, as well as on the temperament and character inventory of personality (TCI-140) and EI. At first testing, they were asked to freely describe three personal goals and to make action plans to achieve each within the next two weeks. Two weeks later they reported on progress on their personal goals (PPG).

**Results**. PPG correlated positively with active procrastination and negatively with passive procrastination. Dividing the participants into median splits on active and passive procrastination resulted in four groups: Active, Passive, Active-Passive, and Non-Procrastinators. Analysis of variance showed that active procrastinators had an advantage in temperament and character traits as well as EI. Active procrastinators were also higher than the other groups on personality profiles i.e. combinations of traits; dependable temperament and well-developed character.

**Conclusions**. Active procrastination can be an adaptive and productive coping style. It is associated with dependable temperament, well-developed character, and high emotional intelligence and predicts meeting personal goals.

## INTRODUCTION

Procrastination has been extensively studied, especially in college students. It is often considered a self-imposed, self-handicapping behavior, and is associated with a variety of personality, situational, psychological and motivational variables (*Steel, 2007*). For example, *Ariely & Wertenbroch (2002)* in a series of experiments showed that students at MIT did not set themselves meaningful and helpful deadlines in order to overcome procrastination and that their self-imposed deadlines did not contribute to the optimization of their academic performance in a semester-long course, concluding that procrastination was a failure of self-control. *Wu et al. (2016)* measured the event related potential of higher and lower procrastinators, finding that high procrastinators preferred immediate rewards

Corresponding author
Ada H. Zohar, adaz@ruppin.ac.il

over delayed and bigger rewards. A study that examined academic procrastination and goal achievement on a weekly basis of web-based protocols (*Wäschle et al., 2014*) found that high procrastinators were low on goal achievement, and in turn, low achievement reinforced academic procrastination forming a positive feedback loop. Procrastination has been described as a failure of self-control (*Pychyl & Flett, 2012*) and as a meta-cognitive failure (*Fernie et al., 2017*).

*Chu & Choi (2005)* suggested a distinction between a non-adaptive type of procrastination "passive", and an adaptive type of procrastination "active". While passive procrastination is a self-destructive process in which self-doubt, anxiety, and distress accompany the non-accomplishment of tasks, and the failure to meet deadlines, active procrastination is a self-regulating time-management strategy that allows working under pressure and meeting deadlines successfully. *Choi & Moran (2009)* proposed and validated an active procrastination scale in a sample of undergraduate college students, which measured four components of active procrastination: Preference for pressure, intentional decision to procrastinate, the ability to meet deadlines, and outcome satisfaction. This active procrastination scale was validated against academic performance. Students high in active procrastination had higher grades, and greater self-reported academic performance. In their study active procrastination, and in particular the ability to meet deadlines, correlated positively with personality traits that confer resilience, conscientiousness and emotional stability (*Choi & Moran, 2009*).

There has been extensive work tying procrastination types with emotional intelligence. Emotional intelligence (EI) was defined by *Salovey & Mayer* (*1990*, page 189) as "the ability to monitor one's own and others' feelings and emotions, to discriminate among them and to use this information to guide one's thinking and actions". If passive procrastination is a failure to guide one's thinking and actions, while active procrastination is a strategy based on one's excellent understanding of one's own motivations and resources, then surely EI would contribute to active and diminish passive procrastination. There is some proof of this hypothesis. EI has been found to be negatively associated with procrastination by employees in the workplace (*Wan, Downey & Stough, 2014*) and in college students trying to meet college requirements (*Deniz, Tras & Aydogan, 2009*). It has also been found to mediate the relationship between procrastination and academic achievement in students with and without learning disabilities (*Hen & Goroshit, 2014*). As *Peña Sarrionandia, Mikolajczak & Gross (2015)* show in their meta-analysis, emotional intelligence is closely linked with emotion regulation, and thus is a trait that is tied to the behavior regulation necessary for avoiding procrastination. Moreover, there is some evidence for a causal relationship between emotional intelligence and procrastination. *Eckert et al. (2016)* showed in a randomized controlled trial, that an intervention enhancing emotional intelligence had a significant effect on reducing procrastination.

*Kim, Fernandez & Terrier (2017)* studied active and passive procrastination in college students. They found that the two were negatively correlated. In terms of personality traits, passive procrastination was positively correlated with neuroticism, and negatively with conscientiousness, while active procrastination showed the reverse pattern of correlations. When personality traits and procrastination scores were used to predict GPA, passive

procrastination negatively contributed to GPA while active procrastination did so positively. Thus passive procrastination is related to neuroticism, while active procrastination is related to resilient personality as well as to greater academic success. Similar findings were reported by *Zhou (2018)* for both male and female college students in a vocational college. *Seo (2012)* found that when both active and passive procrastinators study for the same length of time before an exam, active procrastinators get higher grades than do passive procrastinators. Active procrastination was found to be positively related to creative ideation and creative self-efficacy (*Liu et al., 2017*) and to internal motivation and students' well-being (*Habelrih & Hicks, 2015*). Other studies that compared active and passive procrastination found that, while in both cases a behavioral delay is apparent, the motivational variables underlying active procrastination are completely different from those underlying passive procrastination. *Corkin, Shirley & Lindt (2011)* suggested calling active procrastination active delay. On the other hand, *Chowdhury & Pychyl (2018)* have argued that active procrastination may not be a wholly beneficial or a unitary construct. Like others, they point out that being active and procrastinating are mutually incompatible; *purposeful delay* is the adaptive component of what is commonly referred to as active procrastination. The other component, *arousal delay*, is the need for high arousal which can be experienced by certain individuals only by delaying the execution of tasks until the deadline produces palpable external pressure. This arousal delay is maladaptive and is negatively correlated with personality traits bestowing resilience, while purposeful delay is positively correlated with the same traits.

To date, most research on procrastination and personality used the five-factor-model of personality (*Steel, 2007*). While this model of personality has much to commend it, it is not helpful in distinguishing between earlier developing temperamental tendencies, and later developing, character traits. The temperament and character model of personality (*Cloninger, 2004*) posits that personality is two-tiered: (1) Temperament is present early in development, before language acquisition, and individual differences in temperament are related to individual differences in brain structure and function. Thus temperament traits tend to be pre- or unconscious, and stabilize relatively early in development. (2) The second tier of personality is character, formed later in development, in transaction with the environment, and influenced by the individual's temperament. Character traits are more accessible to the individual, and more susceptible to change (*Cloninger, 2004*). Moreover, character traits are central to self-regulation and self-management. Some support for the earlier development of temperament vs. character is to be found in longitudinal studies (*Zohar et al., 2018*). The temperament and character model of personality is consistent with personality traits not working independently of each other; rather, trait combinations interact to wield influence on the individual's cognitions, feelings and actions (*Cloninger, 2004*; *Cloninger & Zwir, 2018*) forming temperament and character profiles.

According to the temperament and character model of personality (*Cloninger et al., 1994*) there are four temperament traits: Novelty Seeking (NS), an excitatory tendency related to curiosity, exploration, and impulsivity; Harm Avoidance (HA), an inhibitory tendency related to risk aversion, pessimism and anxiety; Reward Dependence (RD) the extent to which an individual is affected by social cues, is sentimental, and shares his

emotional experiences, and Persistence (PS), the temperamental drive that resists the extinction of learned associations and promotes ambition, perfectionism, and toleration of frustration. Individuals who are high in RD and PS, and low in HA and NS, those with a *dependable temperament profile*, tend to be better adjusted, and lead healthier, happier, and more productive lives (*Cloninger & Zohar, 2011*). Thus if active procrastination is a helpful time-management strategy, misnamed as procrastination, it should be associated with a dependable temperament profile, while passive procrastination, a self-destructive and distressing behavior should not.

There are three character traits: Self-Directedness (SD), the character trait that allows individuals to accept themselves, to define personal goals and to mobilize personal resources to work in a directed way in order to achieve these goals; Cooperation (CO), the ability to accept others and to work with them in a principled and equitable way, and Self-Transcendence (ST), the feeling that one is part of a bigger whole, and is open to spiritual experience. Being high in SD and CO is an indication of a *mature personality*, just as being low in these two character traits predisposes individuals to personality disorders (*Cloninger & Svrakić, 2016*). Being high in all three character traits, being of *well-developed character*, is related to happiness, health and health behavior (*Cloninger & Zohar, 2011*) and to greater coherence of heart rate variability (*Zohar, McCraty & Cloninger, 2013*). Individuals with a well-developed character profile are more able to regulate their behavior, in conjunction with their temperament and environmental demands in order to meet their personal goals, while working with others, in the service of a greater good (*Cloninger, 2004*). Thus, if the well-developed character profile is associated with active procrastination, it will provide further proof that active procrastination is not a failure of self-management but rather a successful strategy; while we expect that passive procrastination will not be associated with the well-developed character profile.

The purpose of the current study was to study active and passive procrastination in terms of the bio-psycho-social personality model of temperament and character as well as in terms of emotional intelligence. The study hypotheses were:

(1) There will be a negative association between active and passive procrastination.
(2) Passive procrastination will be associated with less effective goal attainment, while active procrastination will be associated with more effective goal attainment.
(3) Active procrastinators will be higher on EI than Passive procrastinators.
(4) Active and Passive procrastinators will have different levels of personality traits: Active procrastinators will be lower in HA, higher in RD and PS, as well as in SD and CO.
(5) Active Procrastinators will have score higher than the Passive Procrastinators on the dependable temperament profile score.
(6) Active Procrastinators will score higher on the well-developed character profile than the Passive Procrastinators.

## MATERIALS & METHODS

### Participants

The participants were 126 healthy community volunteers of whom 22 (17.4%) were men; their average age was 28.9 ± 9.

## Procedure

The study received the approval of the Ruppin Academic Center Ethics committee 2017-023 L/nd. The study was put on a Qualtrics (Provo, UT, USA) platform, and the first screen provided information on the study, gave contact details of the authors, and required written consent so as to continue to the next screen. Participants were either psychology majors who participated for credit, or individuals who answered a Facebook or WhatsApp invitation with the link to the study. Participants were rewarded for their participation by a raffle of a voucher for a breakfast for two. The study design was short-term longitudinal. At base-line participants reported on the TCI-140, as well as being asked to formulate three goals they wished to achieve within the next two weeks, and to specify their action plan for each goal. Two weeks later, the participants were sent a link in which they were asked to self-report on the TPS, APS, and the EI. At the end of this Self-Report they were presented with the three goals they had formulated two weeks before, and asked to report if they had achieved each of the goals, partially achieved it, or not at all. Participants' privacy and anonymity were protected throughout.

## Instruments

1. Tuckman Procrastination Scale (TPS; *Tuckman, 1991*): The TPS has 16 items that assess procrastination, and are answered on a 7-point Likert-like response scale. Sample items are "I needlessly delay finishing jobs, even when they're important" and "I am an incurable timewaster". The internal reliability of the TPS in this study was $\alpha = 0.95$.

2. Active Procrastination Scale (APS; *Choi & Moran, 2009*). The APS has 16 items that are answered on a 7-point Likert-like response scale. Sample items are "I finish most of my tasks exactly on deadline, because that is how I choose to operate" or "So as to make the maximal use of my time, I intentionally put off some of my tasks". In the current study the internal consistency of the APS was $\alpha = 0.78$.

3. The Temperament and Character Inventory (TCI-140; *Cloninger et al., 1994*): The TCI-140 includes 140 items which are answered on a 5-point Likert-like response scale. The scale includes 20 items for each of the seven traits it measures except for Self-Direction, which has only 16 items, allowing for a 4-item validity scale. The TCI measures four temperament traits: Novelty Seeking (NS) in the current study had internal consistency of $\alpha = 0.64$; Harm Avoidance (HA) had internal consistency of $\alpha = 0.88$; Reward Dependence (RD) had internal consistency of $\alpha = 0.79$, and Persistence had internal consistency of $\alpha = 0.84$. In addition, the TCI measures three character traits. Self-Directedness (SD) had internal consistency of $\alpha = 0.88$; Cooperation (CO) had internal consistency of $\alpha = 0.78$; and Self-Transcendence (ST) had internal consistency of $\alpha = 0.89$. A description of the traits, and of the psychometric properties of the TCI-140 is given in (*Zohar & Cloninger, 2011*).

4. Emotional Intelligence (EI; *Schutte et al., 1998*). The EI includes 33 items which are answered on a 5-point Likert-like response scale. Sample items are "Other people find it easy to trust me" or "I control my emotions". In the current study the EI had internal consistency of $\alpha = 0.89$.
5. Personal Goals. At first measurement, respondents were asked to list three personal goals which they wanted to achieve within the next two weeks. For each of their three stated goals they were asked if they had an action plan (score of 3) a partial plan (score of 2) or no plan at all (score of 1). Two weeks later they were asked to specify on a three-point scale the degree to which they had achieved each of their previously defined goals and could answer that the goal was achieved (score of 3) partially achieved (score of 2) or not achieved (score of 1). Each participant thus could score 3–9 on the degree of planning at Time 1 and the degree of actual procrastination at Time 2 with a potential range of 3–9. The mean score over the three action plans was the participant's Goal Planning (GP) score, while the mean level of goal achievement over the three goals was the participant's progress in personal goals (PPG).

## Data analysis

Data was downloaded from the Qualtrics platform into SPSS files. All analyses were conducted in SPSS 23. Descriptive statistics and correlations were conducted on the continuous variables. In addition, we made a median split of the Passive and Active procrastination scale scores, to produce a "high" and "low" score on each, which allowed for dividing the sample into four groups. A crosstabs procedure with the Chi-square statistic was used to test for an association between high and low active and passive procrastination. An analysis of variance was conducted comparing these four groups for personality traits and EI. Then a dependable temperament profile was formed by adding RD and PS and subtracting HA and NS, thus giving a single temperament score for each participant; and a well-developed character profile was formed using the product of the three character trait scores (Zohar et al., 2018).

## RESULTS

### Descriptive statistics

Table 1 shows the means, standard deviations and inter-correlations of the study variables. The correlations are all in the expected directions, and most of them (those appearing in bold font) are significant at $p < 0.01$. The inter-correlations between the seven traits of the TCI are weak or moderate, as are the other significant correlations in the Table.

Content analysis of the personal goals listed by the participants to be achieved within the next two weeks were highly varied in domain and specificity. The most frequent domain was academic tasks and academic success (30.8%). Examples of academic goals were: "finish all the reading assignments"; "hand in the psychology homework"; "get down to studying". Work related goals constituted 12.7% of the entries. Examples were "Get a raise", "complete 20 placements", and "feel confident in myself, in the next challenge of my dream job". Many participants related to physical activity (11.4%). Some quite specifically "run 12 km". "swim 3 times" but also more generally "exercise" or "start physical activity". The next most frequent domain was body weight (8.7%) "lose weight" or more specifically "lose 2 kilograms". Health behaviors were the next most frequent category (6.7%) and included goals like "smoke less marijuana" "go to the doctor", and "eat a healthy balanced diet and cut down on sweets". Domestic goals constituted 6% and included goals like "go

**Table 1  Descriptives and correlations for study variables (N = 126).**

|      | HA     | RD     | PS     | SD     | CO     | ST    | AP     | PP     | EI     | GP     | PPG    |
|------|--------|--------|--------|--------|--------|-------|--------|--------|--------|--------|--------|
| NS   | −.242  | .127   | −.093  | −.334  | −.148  | .444  | −.019  | .370   | .128   | −.138  | −.265  |
| HA   | –      | −.236  | −.406  | −.494  | −.247  | −.136 | −.512  | .305   | −.490  | −.329  | −.270  |
| RD   | –      | –      | .252   | .257   | .467   | .131  | .185   | −.183  | .336   | .118   | .121   |
| PS   | –      | –      | –      | .399   | .278   | .000  | .388   | −.533  | .377   | .345   | .196   |
| SD   | –      | –      | –      | –      | .544   | −.083 | −.384  | −.631  | .355   | .469   | .406   |
| CO   | –      | –      | –      | –      | –      | .115  | .235   | −.280  | .373   | .124   | .145   |
| ST   | –      | –      | –      | –      | –      | –     | .007   | .075   | .113   | −.044  | −.080  |
| AP   | –      | –      | –      | –      | –      | –     | –      | −.311  | .315   | .244   | .266   |
| PP   | –      | –      | –      | –      | –      | –     | –      | –      | −.412  | −.444  | −.394  |
| EI   | –      | –      | –      | –      | –      | –     | –      | –      | –      | .258   | .329   |
| GP   | –      | –      | –      | –      | –      | –     | –      | –      | –      | –      | .347   |
| Mean | 58.0   | 71.5   | 66.1   | 69.6   | 78.7   | 42.9  | 4.2    | 3.58   | 3.6    | 5.2    | 5.2    |
| (SD) | (11.8) | (9.1)  | (10.9) | (11.4) | (8.6)  | (12.6)| (0.8)  | (1.4)  | (0.3)  | (1.4)  | (1.4)  |

**Notes.**

NS, Novelty Seeking; HA, Harm Avoidance; RD, Reward Dependence; PS, Persistence; SD, Self-Directedness; CO, Cooperation; ST, Self-Transcendence; AP, Active Procrastination; PP, Passive Procrastination; EI, Emotional Intelligence; GP, Goal Planning—Action plans for personal goals; PPG, progress on personal goals.

over the children's rooms" but also "buy house". Another 5% described goals having to do with leisure, such as "rest", or "go abroad". These seven categories together describe 81.9% of the personal goals. In addition, there were some interpersonal goals like "spend uninterrupted quality time with my daughter" or "reconciliation with my friend R". Of the remainder some were too general to classify e.g., "success". Some were highly specific and personal: "when I next visit home remain warm and friendly toward my family even though therapy has revealed some troubling issues" or "attend Rainbow gathering".

*Hypothesis 1: There will be a negative association between active and passive procrastination*

As can be seen in Table 1 there was a negative correlation between the two ($r = −.311$, $p < 0.01$), supporting the negative association hypothesized.

Median splits were made for the TPS and the APS producing high and low passive procrastinators and high and low active procrastinators respectively. Table 2 shows their distribution. Numerically the high–high and low–low group were bigger than the high–low groups along the other diagonal. However, this difference was not statistically significant according to the Chi-square test ($\chi2 = 0.29$, $p > 0.05$). Fully 19.8% of the participants were high on both forms of procrastination, and 23% were low on both. Thus when dichotomized, there is no negative association between high and low active and passive procrastination. This does not support hypothesis 1,

*Hypothesis 2: Passive procrastination will be associated with less effective goal attainment, while active procrastination will be associated with more effective goal attainment.*

The passive procrastination score was moderately and negatively correlated with having formed action plans for personal goals and for progress in personal goals, thus the higher an individual in passive procrastination the less well-formed were his action plans for achieving his personal goals, as well as being less effective in attaining his personal goals.

**Table 2** Distribution of passive and active procrastinators.

| | | Passive procrastination | | |
| --- | --- | --- | --- | --- |
| | | **Low** | **High** | |
| **Active Procrastination** | Low | 29 (23%) | 36 (28.6%) | 65 |
| | High | 36 (28.6%) | 25 (19.8%) | 61 |
| | | 65 | 61 | 126 |

Notes.
Although the High-Low groups are numerically smaller than the High–High and Low–Low groups this difference is not significant, ($\chi^2 = 0.29$, $p > 0.05$), i.e., there is no association between the active and passive procrastination when dichotomized into high and low.

Active procrastination was weakly and positively correlated with formulating action plans and with goal achievement suggesting an advantage for active procrastinators in forming action plans and with achieving their personal goals. These correlations lend support to hypothesis 2.

*Hypotheses 3 & 4: Active procrastinators will be higher on EI than Passive procrastinators. Active and Passive procrastinators will have different levels of personality traits: Active procrastinators will be lower in HA, higher on RD and PS, as well as higher on SD and CO.*

To test these hypotheses we conducted analysis of variance With Bonferroni correction for multiple post-hoc comparisons. To be conservative we included all 4 groups—Active Procrastinators, Passive Procrastinators, Both and Non-Procrastinators.

As is shown in Table 3, hypotheses 3 and 4 were largely supported. Active procrastinators were significantly higher than passive procrastinators on the following traits: Emotional Intelligence, Persistence, Self-Directedness and Cooperation, and significantly lower on Novelty Seeking and Harm Avoidance. Contrary to hypothesis 4, Reward Dependence was no different between the four procrastination groups.

*Hypothesis 5: Active Procrastinators will score higher than Passive Procrastinators on the dependable temperament profile (RD+PS-HA-NS).*

Oneway analysis of variance was conducted for dependable temperament (high Reward Dependence, high Persistence, low Harm Avoidance and low Novelty Seeking) for the four procrastinations groups. The four groups were very different for dependable temperament $F(3, 122) = 19.37$, $p < 0.0001$. Post-hoc comparisons with Bonferroni correction for multiple comparisons showed that passive procrastinators were significantly lower than the other three groups, and active procrastinators were higher than the passive procrastinators and the passive-active procrastinators "both". The means of the four groups + the SE is shown in Fig. 1.

*Hypothesis 6: Active Procrastinators will score higher than Passive Procrastinators on the well-developed character profile (SD × CO × ST).*

Oneway analysis of variance was conducted for well-developed character (high Self-Directedness, high Cooperation, and high Self-Transcendence) for the four procrastination groups. The four groups were significantly different $F(3, 122) = 5.79$, $p < 0.001$. Post-hoc comparisons with Bonferroni correction for multiple comparisons showed that active procrastinators were higher than the passive procrastinators and the non-procrastinators "none". The means of the four groups + the SE is shown in Fig. 2.

**Table 3** Analysis of variance of the four procrastination groups for all individual traits measured.

| Trait | Procrastination group | Mean (SD) | $F(3,122)$ ($p$) |
|---|---|---|---|
| Emotional | Active Procrastinators | 3.8 (0.3)[a] | 8.6 (0.001) |
|  | Passive Procrastinators | 3.4 (0.3) |  |
| Intelligence | Both | 3.7 (0.2) |  |
|  | Non-Procrastinators | 3.6 (0.4) |  |
| Novelty | Active Procrastinators | 53.6 (7.5)[a] | 5.0 (0.003) |
|  | Passive Procrastinators | 58.3 (9.4) |  |
| Seeking | Both | 61.3 (8.2) |  |
|  | Non-Procrastinators | 54.6 (8.8) |  |
| Harm | Active Procrastinators | 51.1 (9.3)[a] | 11.7 (0.001) |
|  | Passive Procrastinators | 64.4 (11.5) |  |
| Avoidance | Both | 54.4 (7.8) |  |
|  | Non-Procrastinators | 61.8 (12.6) |  |
| Reward | Active Procrastinators | 72.1 (8.5) | 2.1 (0.1) |
|  | Passive Procrastinators | 63.4 (10.4) |  |
| Dependence | Both | 72.9 (8.2) |  |
|  | Non-Procrastinators | 73.2 (8.2) |  |
| Persistence | Active Procrastinators | 71.6 (7.6)[a] | 15.8 (0.001) |
|  | Passive Procrastinators | 57.3 (9.9) |  |
|  | Both | 67.4 (10.7) |  |
|  | Non-Procrastinators | 68.9 (9.2) |  |
| Self | Active Procrastinators | 78.2 (9.8)[a] | 14.9 (0.001) |
|  | Passive Procrastinators | 62.8 (11.8) |  |
| Directedness | Both | 68.0 (6.4) |  |
|  | Non-Procrastinators | 68.8 (9.7) |  |
| Cooperation | Active Procrastinators | 82.0 (7.1)[a] | 3.2 (0.02) |
|  | Passive Procrastinators | 76.6 (9.0) |  |
|  | Both | 78.9 (7.7) |  |
|  | Non-Procrastinators | 76.8 (9.3) |  |
| Self | Active Procrastinators | 41.9 (12.8) | 0.3 (0.8) |
|  | Passive Procrastinators | 43.3 (12.0) |  |
| Transcendence | Both | 44.7 (31.1) |  |
|  | Non-Procrastinators | 42.1 (12.8) |  |

**Notes.**
[a] Active procrastinators significantly different from passive procrastinators in post-hoc comparisons with Bonferroni correction.

## DISCUSSION

In the current study the scale scores of passive and active procrastination correlated negatively, as expected, showing that these are contradictory behavioral tendencies. This is consistent with the findings of *Kim, Fernandez & Terrier (2017)* and others. Passive procrastination scores correlated negatively with action plans for personal goals at baseline and on goal achievement two weeks later, while active procrastination had positive correlations. The avoidance that is passive procrastination is non-functional while the strategy of active procrastination is useful in discharging one's goals. This shows the
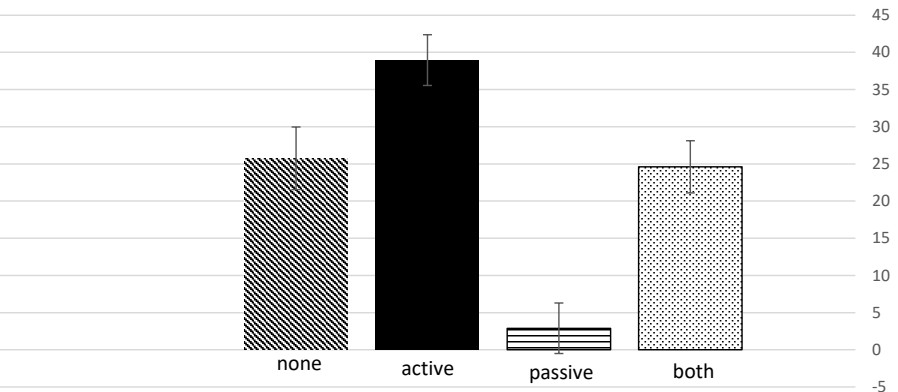

**Figure 1** Dependable temperament scores for the four procrastination groups. .

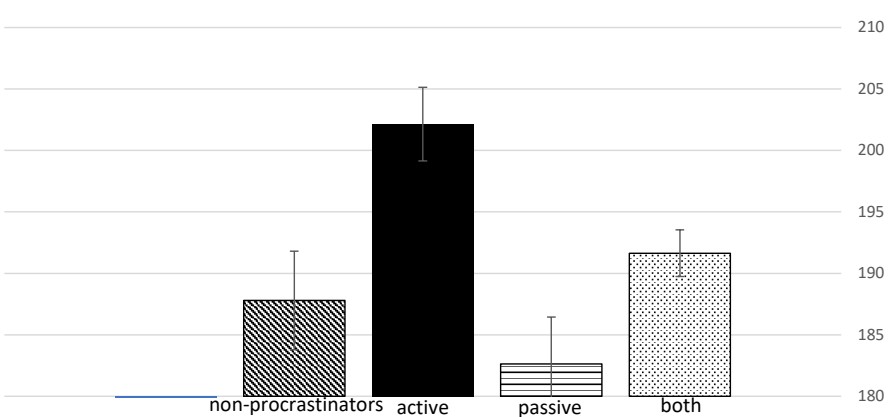

**Figure 2** Well-developed character scores for the four procrastination groups. .

efficacy of active procrastination as has been reported by others (e.g., *Habelrih & Hicks, 2015*).

In the current study we compared four groups formed by dividing the respondents above and below the median for the active and passive procrastination scale scores. This procedure resulted in four groups: those high in passive procrastination and low in active procrastination "passive procrastinators"; Those high in active procrastination and low in passive procrastination "active procrastinators"; Those above the median for both forms of procrastination "active-passive procrastinators"; and those below the median for both "non-procrastinators". The median split of the scales produced four nearly equal groups in number. It is not quite clear how being high on both forms of procrastination might manifest itself. It is possible that these different tendencies are expressed in different ways at different times. Active procrastination has four components (*Choi & Moran, 2009*): (1) a preference for pressure that motivates individuals to delay until the perceived pressure exerted by the encroaching deadline rises above threshold; (2) an intentional decision to delay in discharging a task; (3) an ability to meet deadlines and (4) satisfaction with the
outcome. Unlike passive procrastination, active procrastination is not a unitary concept (*Chowdhury & Pychyl, 2018*), and it is possible that those high in active procrastination are not equally high in all its components. For consistency, we conducted the analyses and comparisons between all four groups.

When each trait was considered separately, active procrastinators were significantly higher than passive procrastinators on: Emotional Intelligence, Persistence, Self-Directedness, and Cooperation. Thus the active procrastinators were more aware of their emotions and those of others and were better able to manage themselves and avoid distress (high EI), they were more ambitious, perfectionistic and frustration-tolerant (high PS), they were more goal-oriented, responsible, resourceful, and more self-accepting (high SD), as well as more empathic, more helpful, and more accepting of others (high CO). Active procrastinators were significantly lower than passive procrastinators on Harm Avoidance and Novelty Seeking, i.e., they were less pessimistic, less fatigable, less shy, and less wary of the unexpected (low HA), as well as less impulsive, more rule-bound, and less explorative (low NS). Thus the two groups were different in the expected directions for six of the eight psychological traits measured in this study. These results are consistent with previous research that found that effective time-management was positively associated with emotional intelligence (*Cerezo et al., 2017*). Emotional intelligence includes the ability to monitor one's own and others feelings and emotions, to discriminate among them and to use them to guide one's actions (*Salovey & Mayer, 1990*). Thus, active procrastinators can keep their motivation and energy harnessed by time-regulation strategies that suit their circumstances as well as their temperament and character. The temperament and character differences between the Active and Passive Procrastinators all favor Active over Passive Procrastinators in that they are associated with greater physical and mental health (*Cloninger & Zohar, 2011*; *Zohar, McCraty & Cloninger, 2013*), happiness (*Cloninger, 2004*), and greater productivity (*Kono, Uji & Matsushima, 2015*).

In addition to comparing the groups for individual traits, we were able to use the temperament and character model of personality (*Cloninger, 2004*) in order to form two personality profiles: (1) the dependable temperament profile, i.e., high in Reward Dependence and Persistence and low in Harm Avoidance and Novelty Seeking. Individuals possessing this combination of temperament trait scores tend to be emotionally and behaviorally well-regulated, sociable (high RD) as well as hard-working and tolerant of frustration (high PS). They neither avoid challenges nor worry over much about the consequence of tackling them (low HA) nor do they tend to act impulsively (low NS). The temperament traits act together in individuals to bring about this harmonious and stable profile (*Zohar et al., 2018*). (2) Well-developed character: individuals high in all three character traits, self-directedness, cooperation, and self-transcendence, are able to weather the demands their temperament and their environment make on them, engaging in self-directed goal-oriented behavior (high SD) working well with others in a respectful way while being empathic and helpful to others (high CO) in order to achieve meaningful goals (high ST). The combination of being high in all three lends the individual resilience and happiness (*Cloninger & Zohar, 2011*). We compared the four groups of procrastinators by means of analysis of variance on these two personality profile. For both, passive

procrastinators were significantly lower than all three other groups and in particular lower than active procrastinators. Thus active procrastinators are more dependable and have better developed character than passive procrastinators.

*Gustavson et al. (2015)* showed that there is considerable genetic influence on procrastination, and that procrastination and impulsivity (a temperamental tendency) have shared genetic variance. In a further study, *Gustavson et al. (2017)* showed that procrastination was genetically related to the executive function of planning and to internalizing and externalizing psychopathology.

Even though temperament and procrastination are genetically influenced they are also subject to environmental influence. *Glick & Orsillo (2015)* tested the efficacy of acceptance-based behavioral therapy for academic procrastination targeting time management. The stated goal of the interventions was to reduce procrastination regarding assigned reading in an undergraduate and a graduate class. *Glick & Orsillo (2015)* failed to bring about an improvement in the participants as a whole, although they did find a significant interaction: students high in academic values improved their procrastination in response to time-management intervention while those low in academic values were not affected. However, *De Paola & Scoppa (2015)*, showed that a simple intervention of setting deadlines improved undergraduates' performance if they were heavy procrastinators. *Hen & Goroshit (2018)* in their editorial, conclude that cost- and time-efficient interventions for procrastination can be deployed in academic settings, and are especially important for learning disordered students.

This study should be read with its limitations in mind. This was a self-selected sample of community volunteers, and included no known extremes of procrastination. All measures were self-reported, including the personal goals, the action plans for goal implementation, and the report of goal attainment. We allowed only two weeks latency between the two measurements, making the window for prediction very short. It is possible that with deadlines that are months or even years in the future, such as those set in graduate school, procrastination of all kinds is not helpful. A bigger random sample, longer latency between T1 and T2, and some objective measures that are not self-reported would constitute improvements. Without extension and replication, it is not clear how generalizable the results and conclusions of this study are.

In conclusion, this study offers additional support for the adaptive aspects of active procrastination. Active procrastination contributed to goal attainment within a two-week deadline, while passive procrastination did not. Participants high in active procrastination were associated with the dependable temperament, and well-developed character and higher emotional intelligence. The results of the current study give further support that what is now often called active procrastination might better be viewed as planned, purposeful time- and self-management strategy.

### Funding

The authors received no funding for this work.

## Competing Interests

Ada H. Zohar is an Academic Editor for PeerJ.

## Author Contributions

- Ada H. Zohar conceived and designed the experiments, analyzed the data, prepared figures and/or tables, authored or reviewed drafts of the paper, approved the final draft.
- Lior Pesah Shimone conceived and designed the experiments, performed the experiments, analyzed the data, prepared figures and/or tables, authored or reviewed drafts of the paper, approved the final draft.
- Meirav Hen authored or reviewed drafts of the paper, approved the final draft, mainly conceptual.

## Human Ethics

The following information was supplied relating to ethical approvals (i.e., approving body and any reference numbers):

Ruppin Academic Center granted Ethical approval to carry out this study (2017-023 L/nd).

## Data Availability

The raw data is available as Supplemental File.

## Supplemental Information

Supplemental information for this article can be found online at http://dx.doi.org/10.7717/peerj.6988#supplemental-information.

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
