# Peer review of "Active and passive procrastination in terms of temperament and character"

_PeerJ, doi:10.7717/peerj.6988_

## Round 0.1 · original submission · Major Revisions

Dear authors

We have now reviewed your ms and both reviewers recommend that we offer you the opportunity to revise and resubmit. I look forward to your revision.

Sincerely
Gerhard Andersson,
Editor

Reviewer 1 ·

Basic reporting

First of all, let me thank you for asking me to review this paper. At the general level it is well written and it addresses most of the literature related to the topic of procrastination. However, the Introduction lacks a consistent logic om how the authors intertwine procrastination-personality-emotional intelligence.

Experimental design

I do think that the research question is well-defined. The authors should develop the limitation section of the paper much further than what is stated in the present version.

Validity of the findings

The findings and conclusions are valid as far as I can see. However, I'm not sure I follow in the rationale about temperament and character profiles. What I can see is that the authors create single scores that then are used to state that, in the case of temperament for example, the higher value means dependable temperament while the lower, the less dependable. I do not see that this is a profile and it does not seem to be used as a profile either. But then again, I might have misunderstood what the authors actually did. In addition, saying that an active procrastinator is creative is a bit problematic...for instance, are all creative individuals active procrastinators? The authors need to adress these type of questions that their findings and design generates.

Reviewer 2 ·

Basic reporting

The English language is at a good level, but it can be improved after the manuscript is improved.
Literature references are satisfactory, with sufficient field background/context provided.
Professional article structure is satisfactory and the article has all the standard sections as well as figures and tables. The data under tables’ and figures’ titles should be transferred to the Results section. Raw data are also shared.
The manuscript includes all results relevant to the hypothesis, but I suggest hypotheses to be better formulated.

Experimental design

This is original primary research within Aims and Scope of the journal.
Research question are not well defined but according to the aim and hypotheses, I assume that they are relevant and meaningful. It is stated how research fills an identified knowledge gap.
The research was performed to a high technical and ethical standard, and methods are described with sufficient detail & information to replicate.

Validity of the findings

My biggest concern is regarding how data were reported in this manuscript and I think there is room for improvement in this part. Once the results are presented in a more understandable and informative manner, the discussion should also be adapted to this. Finally, the conclusions should be better defined and in line with the problems which should be provided.

Additional comments

I have read the manuscript “Active and passive procrastination in terms of temperament and character” several times and I find this to be an interesting research paper that gives a new perspective on procrastination research. I find the research was done properly, but the manuscript in my opinion could be a little better organized and written in the results and the discussion part. Some moderate changes could help this manuscript be more easily understood, more informative regarding the interesting data it presents, and in the end, more easy to read and follow.
As for basic reporting I think that the manuscript is well written, and that the level of English is satisfactory, but there are specific points where some of the sentences could be rephrased (I list them later in specific comments).
As stated above, I think this manuscript has room for improvement, specifically in the relation between aims – results – discussion. It is not quite clear what the aim was because it is described in general, with no specific aims listed. The authors do list the hypotheses, which is good, but the results do not follow those hypotheses clearly and understandably. For some of the analyses it is not clear as to why the authors did them and in what way they help answer their problems, because they do not explain it, and this is not something that will take a lot of time to add. I just think that this should be stated more clearly, because in fact the authors listed 4 hypotheses in the Introduction and then in the Results started answering them in the opposite direction.
You mention 4 hypotheses - I suggest also writing 4 problems first, and then the exact hypotheses – what precisely are you expecting to find regarding those 4 problems, what kind of associations. In the results, after general descriptive part, your result should be presented in a way that makes it clear what you did do answer your problems, one by one, and what the results in this manner show. In the Discussion part you should comment on those results, again in the same order, so it is easy for the reader to follow. An in the end, in Conclusion you should sum up your results as precise answers to your problems stated at the beginning.
All the explanation given under Table or Figure titles should be moved to the Results section.

As for more specific comments, they are listed in a separate file that I will attach.

Annotated reviews are not available for download in order to protect the identity of reviewers who chose to remain anonymous.

---

## Round 0.2 · accepted · Accept

Dear authors

I have now read the response letter and checked the ms and your revisions, and am happy to accept your paper.

Sincerely Gerhard Andersson

#